# Emerging Infectious Diseases Are Virulent Viruses—Are We Prepared? An Overview

**DOI:** 10.3390/microorganisms11112618

**Published:** 2023-10-24

**Authors:** Jasmine J. Han, Hannah A. Song, Sarah L. Pierson, Jane Shen-Gunther, Qingqing Xia

**Affiliations:** 1Division of Gynecologic Oncology, Department of Gynecologic Surgery and Obstetrics, Department of Clinical Investigation, Brooke Army Medical Center, San Antonio, TX 78234, USA; 2Department of Bioengineering, University of California, Los Angeles, CA 90024, USA; hana.song02@gmail.com; 3Department of Clinical Investigation, Brooke Army Medical Center, San Antonio, TX 78234, USA; sarah.l.pierson.mil@health.mil; 4Gynecologic Oncology & Clinical Investigation, Department of Clinical Investigation, Brooke Army Medical Center, San Antonio, TX 78234, USA; jane.shengunther.mil@health.mil

**Keywords:** air surveillance system, avian influenza, emerging virus, Ebola virus, in-air pathogen detection, MERS-CoV, monkeypox, SARS-CoV-2, Zika virus

## Abstract

The recent pandemic caused by SARS-CoV-2 affected the global population, resulting in a significant loss of lives and global economic deterioration. COVID-19 highlighted the importance of public awareness and science-based decision making, and exposed global vulnerabilities in preparedness and response systems. Emerging and re-emerging viral outbreaks are becoming more frequent due to increased international travel and global warming. These viral outbreaks impose serious public health threats and have transformed national strategies for pandemic preparedness with global economic consequences. At the molecular level, viral mutations and variations are constantly thwarting vaccine efficacy, as well as diagnostic, therapeutic, and prevention strategies. Here, we discuss viral infectious diseases that were epidemic and pandemic, currently available treatments, and surveillance measures, along with their limitations.

## 1. Introduction

Throughout human history, viruses have played a complex role. They can lie dormant for extended periods, while in other instances, they can surge forth with astonishing potency to reshape the very course of history. The fragile balance between humans and the microbial world is forever vulnerable to disruption, a lesson engraved into our memory by past devastating outbreaks and pandemics. Since the beginning of the 21st century, several viruses have challenged public health and global communities. The emergence of the Severe Acute Respiratory Syndrome Coronavirus (SARS-CoV) in 2002 [1], followed by the H1N1 influenza outbreak in 2009 [2] and the most recent SARS-CoV-2 pandemic, shook the world and underscored the potential of viruses to swiftly disrupt global health and economies. These events illuminated the concept of “emerging viruses”, those that suddenly breach the barrier between animals and humans, and the challenges they present regarding recognition, containment, and mitigation. In addition, re-emerging viruses, such as avian influenza and coronavirus, showed us their evolutionary powers with renewed vigor. As we stand at the crossroads of scientific discovery and global interconnectivity, the tools we employ to detect and monitor these invisible adversaries play a fundamental role. During this challenge, a Nobel-worthy mRNA vaccine was developed, and the bioaerosol detection method has emerged as a cutting-edge approach that holds the promise of revolutionizing our ability to prevent pandemic infections and identify and track viruses with enhanced precision and efficiency. An overview of the current emerging virulent viruses, preventive methods of vaccination, currently available treatment modalities, and surveillance tools is presented in this study. The information summarized will aid authorities in designing and adopting effective prevention and control strategies to counter the next emerging or re-emerging virus. The viruses reviewed are shown in Figure 1 for their transmission routes and characteristics.

## 2. Emerging and Re-Emerging Viruses

### 2.1. Filoviridae: Ebola Virus and Marburg Virus

Ebola and Marburg viruses are single-stranded RNA filoviruses, which are zoonotic and are often transmitted through direct contact with infected fluids. Both Ebola and Marburg viruses have historically high mortality rates, which have varied from 25% to 90% and 24% to 88%, respectively [3,4]. The largest outbreak of Ebola virus disease (EVD) occurred in West Africa from 2014 to 2016 with more than 28,000 cases [5]. However, both viruses’ MVD case fatality rate is known to be 50%. Both viruses are transmitted through direct contact with infected blood and other bodily fluids. The Ebola virus is believed to have been first contracted by wild animals (such as fruit bats, porcupines, and non-human primates), then it spread to humans, in which human-to-human transmission is possible [3]. The first outbreak of Ebola was recorded in 1976 with an 88% mortality rate [5] and there continue to be outbreaks, mainly in Africa [3]. Marburg virus was first identified in Germany and Serbia in 1967, where laboratory workers were exposed to infected monkeys from Uganda [4].

The wide and fluctuating range of case fatality rate (CFR) data is not truly understood; possible reasons include differences in health status (nutrition, immunity, and co-infection status), genetics (ethnicity-dependent haplotypes or random polymorphisms), health-seeking behavior, case recognition, and the accessibility of supportive care at health-care facilities [5].

### 2.2. Flaviviridae: Zika Virus and Dengue Virus

Zika and Dengue viruses are single-stranded RNA flaviviruses often transmitted by arthropods, such as mosquitos and ticks. Both viruses are transmitted through mosquitos. Although the Zika virus was first identified in 1947 in Uganda, it became more well known when an outbreak occurred in Brazil in 2015, leading to its spread to other parts of the Americas. Before then, it was only sporadically prevalent in Africa and Asia. Its primary concern for public health is how the virus affects pregnancy and newborn babies. A total of 5–15% of infants of women who were infected with Zika while pregnant suffered from Zika-related problems, like the following: microcephaly in the newborn, which is where the baby is born with an underdeveloped head and brain, resulting in neurological disorders and other developmental problems [6]. Dengue has become more prevalent in the past decade, from over 500,000 cases in 2000 to 5.2 million in 2019. Dengue mostly affects Southeast Asia, which accounts for 70% of the global burden. It also seriously affects the Americas and the Western Pacific region. The largest recorded dengue outbreak occurred in 2019, in which the Americas reported 3.1 million cases, and Asian countries, like Bangladesh, Malaysia, Philippines, and Vietnam, recorded many cases as well. Dengue’s severity ranges from being asymptomatic to fatal. The majority of those who contract dengue have mild to no symptoms, in which case they will get better in 1–2 weeks. Common symptoms include high fever, severe headaches, body aches, and rash. Rarely, dengue can become more severe, in which case, symptoms can also include bleeding and even cause death. One who has had dengue in the past is more likely to contract severe dengue the second time [7].

### 2.3. Coronaviruses: MERS-CoV, SARS-CoV, and SARS-CoV-2

MERS-CoV, SARS-CoV, and SARS-CoV-2 are single-stranded RNA coronaviruses and transmitted via airborne particulates. Middle East Respiratory Syndrome (MERS) was first reported in Saudi Arabia in 2012, and all cases of MERS have been traced back to the Middle East. MERS-CoV is first contracted from camels, mainly in the Middle East, Africa, and South Asia, and is then transmitted to humans. However, its specific transmission route between camels and humans still needs to be determined. Due to its limited person-to-person transmission, though possible, it is not considered an epidemic; however, it still poses a global threat because of its high mortality rate [8]. Both SARS-CoV and SARS-CoV-2 originate from China, emerging in 2002 and 2019, respectively. Both viruses caused and continue to cause a global threat due to their high transmission rates [9]. COVID-19, caused by SARS-CoV-2, is still considered a pandemic as it continues to mutate and spread worldwide.

### 2.4. Avian Influenza (H5N1, H7N9)

Avian influenza virus is a strain of the single-stranded RNA influenza A virus that is transmitted between birds through airborne transmission, thus being able to be detected in the form of aerosols. Avian influenza includes the strains H5N1 and H7N9, which are transmissible to humans from birds and considered the most common strains to infect people [10]. Although it cannot be transmitted from human to human, nor is it common for it to be transmitted to humans from birds, it has the potential to become a global threat due to its high fatality rate in humans. With increasing numbers of mammals with avian influenza being detected, it poses risks of the virus being able to infect humans faster and easier since mammals are biologically closer to humans than birds are. Scientists have found mammal-to-mammal airborne transmission possible in ferrets, and only a few mutations take place before it is possible [11]. Scientists have also found a strong possibility of co-infection with avian influenza and the more common human influenza due to the high seroprevalence of both viruses in minks [12]. Thus, the possibility of new mutations arising from co-infection could lead to a greater risk to public health.

### 2.5. Monkeypox Virus

Since May 2022, there have been multiple outbreaks of mpox (previously known as monkeypox) worldwide. It has been declared a public health emergency by the World Health Organization (WHO) as it becomes increasingly transmissible among humans. The monkeypox virus is a zoonotic double-stranded DNA virus in the *Orthopoxvirus* genus, which also includes smallpox. Although its fatality rate is not high, it spreads rapidly, resulting in many hospitalized patients. Mpox was first discovered in Denmark in 1958. Since 2005, mpox outbreaks have been sporadically occurring, mainly in Africa. Thus, the sudden spread of mpox throughout Europe and the Americas caused concern. The monkeypox virus’ primary mode of transmission is through direct contact, particularly affecting men who have sexual contact with other men [13]. A summary of emerging viruses is described in Table 1.

## 3. Vaccination

### 3.1. History

The term vaccine derives from the cowpox virus *Vaccinia* (which comes from the Latin name for cow, *vacca* [40]), which was the earliest laboratory-maintained virus. Variolation was introduced by Dr. Edward Jenner in 1796, who observed that milkmaids previously infected with cowpox showed no symptoms of smallpox after being exposed [41]. Almost 2 centuries after the first smallpox vaccine was introduced, the WHO officially announced the eradication of this disease in 1980 [42]. Sequencing studies have determined that the vaccinia virus is more closely related to horsepox than cowpox, either suggesting an initial misidentification or that subsequent mutations likely accumulated during laboratory propagation. This continued to alter the virus to the extent that vaccinia is no longer known to be a naturally occurring strain [43].

### 3.2. Vaccination, Herd Immunity, and Public Awareness

The widely practiced childhood vaccination program provided immunity before children were exposed to potentially life-threatening diseases. In 2000, measles was declared eliminated in the United States, with a high vaccination rate that resulted in societal herd immunity [44]. However, the decreased vaccination rate driven by misinformation about measles and measles, mumps, and rubella (MMR) vaccines resulted in outbreaks in 2014 with 667 cases and in 2019 with 1274 cases [45]. The outbreaks were transmitted by unvaccinated international travelers acquiring measles abroad and returning to low-vaccinated close-knit vulnerable communities. High coverage with the MMR vaccine is the most effective strategy to limit transmission and obtain herd immunity to maintain the elimination of measles in the United States [46].

Since 2000, multiple outbreaks of vaccine-derived poliovirus (VDPV) have been recorded worldwide in unvaccinated communities [47]. VDPV is a strain related to the weakened live poliovirus contained in the oral polio vaccine (OPV). OPV is a live attenuated combination of three poliovirus serotypes and may have borne out the risk of reversion or gain-of-function mutation if allowed to circulate in under- or unimmunized populations for long enough [48]. Studies show that VDPV type 2 has the highest risk for paralytic disease [49]. In July 2022, an unvaccinated individual from Rockland County, New York, was infected by VDPV type 2. VDPVs emerge when insufficient individuals are vaccinated against polio, and the weakened strain of the poliovirus from OPV spreads among under-immunized populations, causing gain-of-function mutation. The United States has used the inactivated poliovirus vaccine (IPV) since 2000 and protects against paralytic diseases caused by all types of polioviruses, including VDPV. Thus, increasing the population vaccination rate to reach herd immunity is paramount in primary prevention to eradicate polio [48].

### 3.3. Vaccine Hesitancy

Almost counterintuitively, the success of modern vaccines at reducing the morbidity and mortality of once-common illnesses has led to periodic downturns in segments of the public’s willingness to accept vaccines. Despite decades of data from controlled studies collectively supporting the efficacy and benefit of vaccination, small but dedicated movements continue to resist or outright refuse vaccination. Historically, reluctance to vaccinate can be attributed to one of three general biases: religious/moral; cultural; or social/political objections. Even then, the degree of hesitancy tends to lie somewhere along a continuum and can be a complex mix of personal and outside reasons [49]. In the most recent pandemic period, the amplification of messaging using traditional and social media platforms has increased the proportion of those identifying as anti-vax for social and political reasons. Shifting attitudes with respect to government control, collective public health and safety, trustworthiness of science, and the medical profession can all contribute to fluctuations in the willingness to vaccinate.

In a self-fulfilling manner, the reduction in vaccine coverage caused by hesitancy has led to failures in protective immunity. The primary example is the effort launched by the World Health Assembly in 1988 to eradicate poliovirus. At the outset, there was every reason for optimism; smallpox had been declared eradicated in 1977 after a massive global vaccination campaign. There was an effective vaccine readily available for deployment, and teams of medical personnel were committed to the effort. Polio, like smallpox, is an exclusively human disease; there is no other natural reservoir from which a re-introduction could occur. However, twenty-five years on, the global community now declares individual countries as polio-free and regularly endures outbreaks of the disease due to hesitancy but also to regional political and resource instability. Without a widespread collective memory of how destructive some infectious diseases can be, the willingness of individuals to participate for the wider benefit is no longer a certainty.

### 3.4. Vaccine Development in Emerging Viruses

The COVID-19 vaccine was first available to the public on 11 December 2020, under emergency use authorization (EUA), one year after the identification of the SARS-CoV-2 virus in China. This nucleic acid-based mRNA technology yielded the Pfizer/BioNTech COVID-19 vaccine Comirnaty, which received U.S. Food and Drug Administration (FDA) approval on 23 August 2021 [50].

Moderna’s Spikevax, also an mRNA vaccine, received FDA approval in 2022. On the contrary, the Johnson & Johnson COVID-19 (Ad26.COV2.S) vaccine is an adenovirus-based vaccine and was deployed with success under EUA during the declared public health emergency. This adenovirus type 26 is a naturally occurring virus with the replication gene deleted, so it is incapable of replicating within humans but expresses the SARS-CoV-2 spike antigen to induce an immune response and produce the antibodies to protect from future infection. However, clinical trial results in the United States showed the Pfizer and Moderna vaccines to be 95% effective and the Johnson & Johnson vaccine to be 66% effective in protecting against moderate and symptomatic SARS-CoV-2 infection [51].

Multiple Ebola vaccine candidates have entered phase I-III clinical trials since the major West African outbreak in 2013. More than 10 of these were live-attenuated constructs based on the backbone of the vesicular stomatitis virus, VSV [5,52]. The FDA-approved ERVEBO^®^ vaccine, which is a recombinant virus vaccine against Ebola virus disease (EVD), will begin phase 4 trials in 2024, [53]. Several Marburg vaccine trials are underway, but none are approved. Currently, a phase 2 double-blind, placebo-controlled trial using a monovalent chimpanzee adenoviral-vectored Marburg virus vaccine is open for recruitment [54]. For Zika virus, there is no approved vaccine available, but currently, there is a phase 2 trial of the purified inactivated Zika virus vaccine (PIZV) in healthy participants [55].

The most common avian influenza strains that are transmitted to humans are H5N1 and H7N9 viruses [56]. After the emergence of the H5N1 virus in poultry that eventually infected people in China in 2013, it became of great importance to find a vaccine for avian influenza viruses and begin to vaccinate poultry. Due to this, by 2018, China was able to successfully eliminate the spread and infection of the H7N9 avian influenza virus in humans by vaccinating its poultry [57]. In 2007, the first H5N1 avian influenza vaccine was FDA-approved [58], but there is no vaccine available for H7N9. Seasonal influenza vaccines consist of two strains of influenza type A viruses and two strains of influenza type B viruses [59].

## 4. Clinical Manifestation and Treatment

### 4.1. Filoviridae: Ebolavirus and Marburg Virus

Ebola virus disease (EVD) develops abruptly with fever and chills after 6–12 days after exposure [60]. During the incubation period, the infected individual is not contagious to others. However, all symptomatic individuals are assumed to have the virus in the blood and other body fluids. Initial common symptoms include fever, fatigue, headache, and a loss of appetite [61]. The patient can also develop a diffuse nonpruritic rash by day 5 to 7. Diarrhea and vomiting are very common and usually develop within the first few days of illness, which may result in severe fluid loss, leading to dehydration, hypotensive shock, and multi-organ failure [62]. Bleeding can also occur, mostly manifesting as blood in the stool (about 6%), and significant hemorrhage may be seen in the terminal phase of the illness [63].

Marburg virus disease has symptoms like EVD but a higher fatality rate of 80–90% [64]. The symptoms begin abruptly with high fever, severe headache, and malaise. Diarrhea and vomiting can begin on the third day. Many patients develop severe hemorrhage within seven days [65] due to systemic inflammatory response triggering coagulation cascades, leading to disseminated intravascular coagulation [41]. In severe cases, death usually occurs 8 to 9 days after the onset of symptoms, preceded by shock and severe hemorrhage [66].

Management and treatment for the filovirus infections of EVD and Marburg virus disease heavily rely on aggressive supportive care to prevent the development of shock and to maintain adequate organ functions while the host immune system mobilizes an adaptive response to eliminate the virus [67,68]. For Ebola-specific therapies, there are the triple-monoclonal antibody (mAb) REGN-EB3 made of atoltivimab, maftivimab, and odesivimab (Inmazeb) and the monoclonal antibody ansuvimab (Ebanga). In late 2020, the U.S. Food and Drug Administration (FDA) approved these therapies [69,70], which can be used for adult and pediatric patients and are administered as a single dose. On the contrary, there is no approved treatment for Marburg virus disease. mAb therapy against the ebolavirus is ineffective for the Marburg virus because the therapy consists of mAb against the surface glycoprotein of the specific ebolavirus.

### 4.2. Zika Virus

Zika virus infection presents with mild clinical disease and is mostly asymptomatic. Symptoms of Zika virus infection include low-grade fever; pruritic rash on the face, trunk, extremities, palms, and soles; arthralgia; and conjunctivitis [71,72], with other common viral symptoms of myalgia and headache after the incubation period of 2–14 days from the mosquito bite [73]. Although the initial symptoms may be mild, there is a strong association between Guillain–Barré syndrome [74] and other neurological disorders, such as optic nerve damage caused by increased intraocular pressure, which may result in glaucoma [75]. Moreover, maternal infection during pregnancy results in congenital microcephaly, stillbirth, and miscarriages [76,77,78]. This maternal-to-fetal vertical transmission of Zika virus was 65% or even higher in a cohort study of 130 infants because not all infants performed serial testing [79]. This vertical transmission has serious fetal complications of fetal growth restriction, extremity abnormalities [80], and a plethora of central nervous system sequelae, including but not limited to ventriculomegaly, microcephaly, and intracranial calcifications [81].

Currently, the treatment of the infection is still supportive, although researchers developing an antiviral drug have made enormous efforts. Management consists of rest and symptomatic treatment, including fluids to prevent dehydration and the administration of acetaminophen to relieve fever and pain, but avoiding aspirin and other nonsteroidal anti-inflammatory drugs until dengue infection can be ruled out to reduce the risk of hemorrhage [82].

### 4.3. Coronavirus: MERS-CoV, SARS-CoV, and SARS-CoV-2 (COVID-19)

#### 4.3.1. MERS-CoV

Middle East respiratory syndrome coronavirus (MERS-CoV) has an estimated incubation period of 9 to 12 days [83] with common viral infection symptoms of fever and chills with cough and shortness of breath. Most reported patients were severe cases of pneumonia, acute respiratory distress syndrome requiring mechanical ventilation, and acute kidney injury. MERS-CoV infection treatment is supportive, and the World Health Organization provides guidelines for severe acute respiratory infection (SARI) when MERS-CoV is suspected. Early recognition of SARI and the immediate implementation of infection prevention with respiratory support and shock management are critical for effective treatment [84].

#### 4.3.2. SARS-CoV

Severe acute respiratory syndrome (SARS), caused by the SARS-associated coronavirus (SARS-CoV) outbreak in 2003, was treated with supportive measures, and about 90% (7322/8096) of patients survived. However, half of the patients who required mechanical ventilation died [85]. This global SARS epidemic was contained, and there have been no known cases of SARS since 2004.

#### 4.3.3. SARS-CoV-2

The incubation period for COVID-19 is 14 days after exposure, with most cases occurring after 4 to 5 days [86]. Among symptomatic COVID-19 patients, cough, myalgias, and headache are the most reported symptoms [87]. Other symptoms are low-grade fever, cough, smell and taste abnormalities, rash, and conjunctivitis. Loss of smell was less common in patients infected during Omicron’s prevalence, supporting the idea that patients with recent variants are at a lower risk of developing associated chemosensory loss [88,89]. In early studies, respiratory failure due to acute respiratory distress syndrome developed in 20% after 8 days of symptom onset, and 12.3% of patients had mechanical ventilation [90]. Thromboembolic complication is common, with a reported 31% incidence in ICU patients [91].

The treatment recommendation for COVID-19 disease caused by SARS-CoV-2 has evolved as many randomized controlled trials have been completed. The current guideline by WHO recommends treatments based on a disease severity classification of non-severe, severe, or critical based on respiratory status and oxygen saturation. Based on the WHO severity criteria, treatments of Remdesivir, Nirmatrelvir, corticosteroids, IL-6 receptor blockers, and Baricitinib are recommended [92].

### 4.4. Avian Influenza (H5N1, H7N9)

#### 4.4.1. H5N1

Prior H5N1 infections in humans remain very rare, involving predominantly children and young adults, but they can cause severe disease with a high mortality rate of up to 50% [93]. The most common clinical symptoms are similar to other viral infections, including fever, cough, and dyspnea. Risk factors for severe disease include the presence of bilateral pneumonia. Neutropenia and increased liver enzyme of ALT predicted fatal outcomes [94].

#### 4.4.2. H7N9

Patients with H7N9 infection would show symptoms of fever, cough, dyspnea, headache, myalgia, and malaise, causing severe illness, including pneumonia and ARDS, with a high 78% ICU admission rate and a 28% death rate [95]. Most patients with H7N9 are severely ill, but mild and moderate cases have been reported [96]. For outpatients without fever and symptoms, close monitoring is recommended for the progression of the illness.

For any influenza treatment, oral oseltamivir is recommended, and the peramivir intravenous route is recommended if the patient cannot tolerate the oral therapy. These medications are in the neuraminidase inhibitor category. These therapies have proven efficacy when started early in the diagnosis. However, the emergence of antiviral resistance was also noted, which should be considered in pandemic preparedness planning [97].

### 4.5. Monkeypox Virus (Mpox)

Monkeypox typically presents with systemic symptoms attributable to a viremic phase of illness lasting 1 to 5 days with fever, headache, myalgia, fatigue, sore throat, and back pain after the incubation period from 5 to 13 days. However, during the recent outbreak in May 2022, some patients presented with genital, anal, and/or oral lesions without systemic illness. Proctitis and tonsillitis were more common during the recent outbreak than previously reported [98]. Also, ocular involvement is rare but requires urgent treatment since corneal scarring and vision loss are potential complications [99]. In addition, there is evidence of severe neurological complications of encephalitis and seizure [100]. Most patients with mpox will recover without medical intervention, but antiviral therapy is recommended in severe disease. Currently, tecovirimat is available for mpox treatment, which was approved for smallpox [101]. In addition to tecovirimat, trifluridine can be used as eye drops or ointment for ocular lesions [102]. 

## 5. Surveillance Tools

### 5.1. Air Surveillance System

The COVID-19 pandemic highlighted the importance of science-based decision making and exposed global vulnerabilities in preparedness and response systems. One of the biggest challenges faced during the COVID-19 pandemic has been the lack of an effective surveillance system that can provide early warnings about outbreaks. The development and usage of a vaccine have been proven effective at delivering valid protection to individuals; however, the threat of the appearance of a new variant is constantly challenging the vaccine’s effectiveness. Under these circumstances, the means that can prepare us for the next infectious disease challenge become critical. More importantly, where we should conduct surveillance for emerging diseases is another fundamental question regarding how fast we can detect the source of potential outbreaks.

Robust evidence has shown that inhaling bioaerosol particles containing viable respiratory viruses is the primary route of COVID-19 human-to-human transmission [103,104,105]. Viruses can become airborne through various mechanisms such as talking, sneezing, coughing, the breathing of an infected individual, medical procedures, and flushing a toilet containing infectious particles. Once the viral-laden aerosols (<100 µm) are generated, they can linger in the air for hours and travel beyond 1 to 2 m from the source, causing new infections at both short and long ranges [100]. Currently, we are facing not only the threat from the re-emergence of SARS-CoV-2 but also other respiratory viruses such as Middle East respiratory syndrome coronavirus (MERS-CoV), respiratory syncytial virus (RSV), and influenza. Effectively identifying the viral particles in bioaerosols will be the key to spotting the location of the next novel respiratory virus threat. Since the COVID pandemic started in 2019, the number of studies regarding bioaerosol monitoring has been rapidly increasing, with most studies focused on SARS-CoV-2 in hospital settings. The fundamental step for in-air pathogen detection is the collection of bioaerosols. Several air biosensor collection methods are widely used by researchers: sedimentation, filters, impactors, cyclones, impingers, electrostatic precipitators, and microfluid platforms [106]. Each method can collect bioaerosols from air samples. Furthermore, researchers should carefully choose the method based on the specific research aim or application. The advantages and disadvantages of the commonly used aerosol collection methods are summarized in Table 2.

Bioaerosol particles not only contain pathogenic and/or non-pathogenic live/dead microorganisms but also contain environmental components, including dust, droplets, salt, and other particles [118]. These non-pathogenic components can potentially affect the effectiveness of airborne microorganisms’ collection, isolation, and purification. In addition, the concentration of the virus in bioaerosol particles, room humidity, room temperature, room size, and airflow rate all play a role in air sample collection and virus isolation [119].

### 5.2. Viral Detection Methods

Detecting viruses in air samples can be challenging due to the small size of viral particles and relatively low viral load. In general, all available virus detection methods can be used in bioaerosol samples after elution. The most common in-air viral detection method is the polymerase chain reaction (PCR). This includes the regular PCR to detect DNA viruses, reverse-transcription (RT) PCR to detect RNA viruses, real-time PCR (qPCR) to quantify viruses, and digital PCR to precisely identify the copy number variations. As a widely used molecular DNA and RNA amplification method, the PCR offers many detection advantages, such as sensitivity (especially digital PCR), specificity, and speed (results are usually available within hours). The PCR method can also be automated, simultaneously allowing the high-throughput testing of large-scale samples. On the other hand, the major limitation of the PCR method in emerging virus detection is that this technique requires prior knowledge of the target sequence in the viral genome. Thus, this method cannot be used for unknown viruses. In addition, if the target sequence mutates significantly, the PCR may not be able to detect the new strains that diverged from the target virus. Other than PCR, other virus detection methods include virus culture [120], loop-mediated isothermal amplification (LAMP) [121], enzyme-linked immunosorbent assay (ELISA) [122], immunofluorescence assay (IFA) [123], and next-generation sequencing (NGS) [124]. Due to the ability to sequence genetic material from samples, NGS is widely used to identify unknown viruses [125].

### 5.3. Potential Surveillance Location Identification

Many virus outbreaks of emerging infections originate in animals. These zoonotic viruses include SARS-CoV-2, HIV, Zika, MERS, Ebola, and monkeypox. Humans can acquire zoonotic viruses in various settings where there is close contact between humans and animals, especially wild animals. Although not all interactions between humans and animals lead to virus transmission, certain conditions, such as overcrowding, poor sanitation, close contact between different animal species and humans, and inadequate biosafety measures, can increase the risk of virus spillover. Thus, to reduce the risk of zoonotic virus transmission, environments where human–animal contact occurs may require more frequent monitoring or constant surveillance. Here, we list some places that can act as high-risk disease transmission environments.

A wildlife market/wet market, where live animals or wild animals are sold for food, can be a typical example of this environment. These markets often involve a mix of animal species from different regions, which provide opportunities for zoonotic viruses to cross species barriers and facilitate the transmission of viruses from animals to humans. In December 2019, the first case of COVID-19 was identified in a seafood market in Wuhan, China; then, the virus spread worldwide. According to the World Health Organization (WHO), as of August 2023, over 6.9 million deaths were due to COVID-19 [126].

Farms where animals are raised for food production, including poultry, pigs, and other livestock, can be breeding grounds for viruses. The H1N1 influenza (swine flu) pandemic in 2009 is one example. This novel strain of the H1N1 influenza virus contained genetic elements from pigs, birds, and human influenza viruses. The virus was initially identified in pigs, particularly in North America, but it quickly spread to humans. The close contact between humans and pigs on pig farms played a role in the transmission of the virus to humans. The 2009 H1N1 pandemic that was first detected in California, United States [127], resulted in widespread illness and affected countries around the world.

Tropical and subtropical regions with warm and humid climates are often connected with a high risk of vector-borne viruses. Mosquitos thrive in this environment, increasing the risk of transmitting diseases like Zika, dengue, West Nile fever, and chikungunya [128]. Forests and rural areas have high biodiversity and host a variety of animal species, and their associated vectors can pose an increased risk of virus spillover to humans.

## 6. Discussion

Historically, virus outbreaks have initiated the public response to vaccine development and clinical trials for treatment. The recent COVID-19 pandemic and the recent mpox outbreak showcased the possible effects of constantly changing viruses on society. A genomic mutation develops a new deadly variant by changing its route of transmission and becoming highly infectious with a low viral load. However, our ability to anticipate, respond to, and mitigate the impacts of emerging and re-emerging viruses is more promising than ever due to scientific and medical advancements in this modern era. In addition, with technological advancements in bioinformatics, a rapid, streamlined means of sequencing newly emerging viruses is possible through an automated workflow pipeline [129]. This will facilitate the expedited identification of emerging viruses, resulting in earlier public and medical responses to confine their outbreak. The continuously evolving mutation of viruses may challenge the effectiveness of available vaccines.

In addition, non-invasive environmental surveillance tools for early detection have been developed that are currently used by either research groups or the government, such as wastewater monitoring [130,131] and vector sampling [132]. This method was supported by the provided evidence that non-water-borne viruses can be detected during wastewater monitoring. For example, airborne viruses (coronavirus [121] and influenza [133]) and vector-borne viruses (Zika virus [134] and West Nile virus [135]) were detected via wastewater-based epidemiological studies.

However, whether non-airborne viruses can be detected via bioaerosol monitoring is unclear. Since virus-containing aerosols can be generated through various mechanisms, such as respiratory activities (talking, coughing, and sneezing), medical procedures (intubation, dental procedures, and bronchoscopy), environmental disturbance (vacuuming, cleaning, and sweeping), animal activities, and industrial processes, bioaerosol surveillance can be considered a powerful tool to screen viruses that linger in the air, especially indoor air. Admittedly, this method has some limitations; for example, the potential for a virus to become aerosolized depends on various factors, the virus detection rate can be very low, and some viruses are more capable of remaining infectious in aerosols and surviving airborne conditions than others. In the era characterized by the swift advancement of science and technology, technological limitations are expected to be addressed soon.

Under these circumstances, the means that can prepare us for the next infectious disease challenge become critical. International collaboration, data sharing, and interdisciplinary research are key to staying ahead of viral threats. To prepare for the next outbreak in global public health, research and public effort ought to focus on the availability of reliable surveillance systems for emerging threats, expedited vaccine development as the viral infection emerges and becomes epidemic, public awareness and receptivity towards the benefits of mass vaccination, and practicing good citizenship surrounding the public health measures of testing and quarantines. The information summarized in this review will aid authorities in designing and adopting effective prevention and control strategies to counter the next emerging or re-emerging virus.

The topic that we summarized is an overview of a quickly changing field that expands exponentially with new research findings. Here, we attempted to provide the current research topics and past knowledge of emerging and re-emerging viruses for readers to further explore their interested field of study. Thus, the limitation of this approach lies in the lack of an in-depth description of each topic and virus discussed.

## 7. Conclusions

COVID-19 disease highlighted the importance of science-based decision making and exposed global vulnerabilities in the prevention and preparedness of pandemic infection and response systems. During the pandemic, medical advancements in developing a new vaccine in a timely manner and improved treatment methods becoming available for viral infections were successes. The development and usage of a vaccine have been proven effective at providing valid protection to individuals, but public awareness and receptivity towards mass vaccination are a challenge. Furthermore, the threat of new variants is constantly challenging the vaccine’s effectiveness. One of the major challenges faced during the last pandemic was the lack of an effective surveillance system that can provide early warnings about outbreaks. From past experiences, we learned that proactive surveillance, rapid diagnostics, and effective communication networks are pivotal in containing and managing emerging threats. In a world bound by shared vulnerabilities, our collective action and unwavering determination hold the key to safeguarding future generations against the threats that emerge from the virus realm. As we peer into the future, preparedness is not an option but a mandate.

## Figures and Tables

**Figure 1 microorganisms-11-02618-f001:**
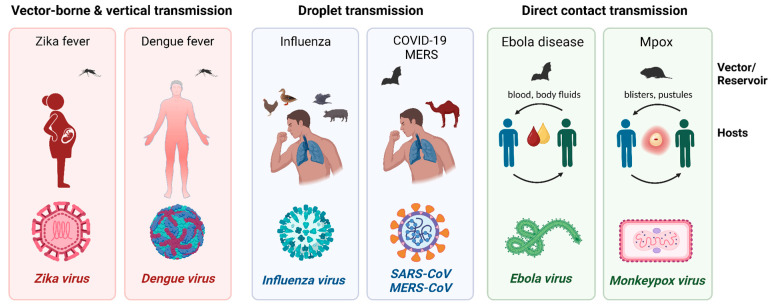
(Re)-emerging viral diseases: vectors, reservoirs, and routes of transmission. The causative viral agents, primary modes of transmission by vectors, reservoirs, and infectious biological materials, and hosts are shown (figure created with BioRender.com).

**Table 1 microorganisms-11-02618-t001:** A summary of emerging viruses.

Virus	Size (Diameter)	Shape	Geo-Regions	Detection Method	References
Ebola	80 nm, varying length	rod-shaped	17 countries (mainly Africa)	RT-PCR, LFA, NGS, antigen test	[3,14,15,16,17,18,19,20,21,22]
Marburg	<14,000 nm	rod-shaped	11 countries (mainly Africa)	RT-PCR, ELISA	[23,24]
Zika	40–43 nm	sphere	89 countries	RT-PCR, ELISA, RT-LAMP	[25,26,27]
Dengue	40–50 nm	sphere	>100 countries	RT-PCR, ELISA, immunoassay	[7,28,29,30]
MERS-CoV	80–120 nm	sphere	27 countries	RT-PCR, CRISPR, biosensor	[8,31,32,33]
SARS-CoV	80–120 nm	sphere	China and four other countries	RT-PCR, CRISPR	[9,31,32]
SARS-CoV-2	80–120 nm	sphere	worldwide	RT-PCR, CRISPR, ELISA, LAMP, LFA, NGS, antigen/antibody test	[31,32,33,34,35]
Avian Influenza	100 nm	sphere	worldwide	RT-PCR	[36,37]
Monkeypox	200–250 nm (length)	brick-shaped	110 countries	RT-PCR	[13,38,39]

RT-PCR, real-time polymerase chain reaction; LFA, lateral flow assay; NGS, next-generation sequencing; ELISA, enzyme-linked immunosorbent assay; RT-LAMP, reverse-transcription loop-mediated isothermal amplification, CRISPR, clustered regularly interspaced short palindromic repeats.

**Table 2 microorganisms-11-02618-t002:** A summary of the commonly used methods for bioaerosol collection.

Sampling Method	Collection Mechanism	Advantage	Disadvantage	References
Natural sedimentation	Gravity. Collect the settling bioaerosol using a nutrient agar plate or swabs	Low cost, easy operation, and little impact on microbial activity	Low efficiency; increases microbial risk resulting from cultured pathogenic microorganisms	[106,107]
Filtration	Bioaerosol collected on filter media through interception, impaction, and diffusion	High collection efficiency, low cost, and easy to operate	Easily blocked, low collection velocity due to the fragility of the filters, and the air environment may increase the sampling difficulty	[106,108]
Centrifugation/cyclone	Centrifugal force deviates bioaerosol into the collection wall or liquid	Compact size, continuous-flow collection, and ability to collect the virus in different particle sizes	Low collection efficiency for small bioaerosols (<1 µm), virus deactivation upon collection, and evaporation of the thin liquid film	[109,110]
Impaction	Bioaerosol first drawn into a nozzle with a vacuum pump, then impacted onto a solid collection media	Cost-effective and easy to use	Reduced collection efficiency due to the deposition of the bioaerosol on the impactor wall; decreased bioactivity due to the shear force on the particles	[103,111]
Impingement	Bioaerosol is sucked into a chamber through a nozzle and captured with liquid collection media	The virus can be detected without the elution process	Reduced viability due to the shear forces in the nozzle; particles adhere to the wall of the collection chamber	[112,113]
Electrostatic precipitators (ESPs)	Bioaerosols are charged by metal needles at the inlet of the ESP, then travel in the direction of collecting electrodes	High collection efficiency can enrich viral particles by more than 10^6^-fold	Extra preparation steps for additional material are needed for viral enrichment	[114,115]
Microfluidics	Relies on differently structured microfluid chips to trap and concentrate particles	Low cost, easy integration, and automatic operation	Small sampling volume	[116,117]

## Data Availability

Not applicable.

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
