# Peer review of "Emerging Infectious Diseases Are Virulent Viruses—Are We Prepared? An Overview"

_microorganisms, 2023, doi:10.3390/microorganisms11112618_

Round 1

Reviewer 1 Report

This review is a brief description of several recent viral pandemics, some information about those viruses, currently recommended and available  treatments, and some information on general methods of surveillance. Overall, the review highlights the important issues revealed during recent viral disease outbreaks but it lacks rigorous analysis of published information and lack of references to scientific publications.

There are several corrections required:

1. Lines 50-57: The information about the mortality rates for Ebola virus is not accurate and up to date. It would be very useful to mention the differences observed during different outbreaks due to the way infections were spread.  Check even the WHO sites about EBOV disease. Please add references to that part

2.    Table 1 is missing information about MERS-CoV and SARS-Cov-2 which is easily available from scientific publications.

Author Response

There are several corrections required:

  1. Lines 50-57: The information about the mortality rates for Ebola virus is not accurate and up to date. It would be very useful to mention the differences observed during different outbreaks due to the way infections were spread.  Check even the WHO sites about EBOV disease. Please add references to that part

We greatly appreciate the input and suggestions to make this manuscript stronger and scientifically relevant to the readers.

The correction is made. The information provided from WHO sites was confirmed. New reference was added to explain the differences in mortality rate in different outbreaks.

  1. Table 1 is missing information about MERS-CoV and SARS-Cov-2 which is easily available from scientific publications.

The missing information is added. Dengue virus was also added per another reviewer’s response.

Reviewer 2 Report

The present manuscript authored by Han and collaborators undertook about virulent and infectious viruses that were epidemic and pandemic. Additionally, it explores current available treatments and surveillance methods, along with their inherent limitations. In my assessment, the obtained data lacks compelling elements in terms of both relevance and originality. The resulting data exhibit a surface-level exposition and lack the depth required to fortify the thematic update. Consequently, this work is confined in its capacity to engage in a meaningful discourse. As it stands, the manuscript falls short in delivering the necessary scientific rigor.

Author Response

The present manuscript authored by Han and collaborators undertook about virulent and infectious viruses that were epidemic and pandemic. Additionally, it explores current available treatments and surveillance methods, along with their inherent limitations. In my assessment, the obtained data lacks compelling elements in terms of both relevance and originality. The resulting data exhibit a surface-level exposition and lack the depth required to fortify the thematic update. Consequently, this work is confined in its capacity to engage in a meaningful discourse. As it stands, the manuscript falls short in delivering the necessary scientific rigor.

We greatly appreciate the input and suggestions to make this manuscript stronger and scientifically relevant to the readers.

The manuscript had a major revision to include vaccination section for primary prevention therapy with current on-going trials. We added a table to summarize the detection methods. We also added a figure to show the transmission routes of each virus and the clinical characteristics.

We expanded the discussion section.

The limitation of the paper is revealed in end of discussion section Line 494-498.

The topic that we summarized is an overview of a quickly changing field that expands exponentially with new research findings. Here, we attempted to provide the current research topics and past knowledge of emerging and re-emerging viruses for the readers to search deeper in their interested field of study. Thus, the limitation of this approach results in a lack of an in-depth description of each topic and virus discussed.

Reviewer 3 Report

Your review is very interesting and provides a general overview of the situation.

Further reinforce the importance of these viruses with annual incidence data, drugs in development, or lack thereof to control the infection.

A figure is missing with the transmission routes of each virus and the main clinical characteristics of each infection

A small section of DENV is missing

Line 133-140 letters are shown in gray

Author Response

Your review is very interesting and provides a general overview of the situation.

Further reinforce the importance of these viruses with annual incidence data, drugs in development, or lack thereof to control the infection.

We greatly appreciate the input and suggestions to make this manuscript stronger and scientifically relevant to the readers.

  • The historical largest outbreak incidence rate of Ebola virus is added. The annual incidence rate changes from year to year as the virus emerges and re-emerges, so not all data was added to the manuscript. Vaccination section is added for primary prevention therapy in infection control and additional clinical trials that are currently available are added to the manuscript.

A figure is missing with the transmission routes of each virus and the main clinical characteristics of each infection

  • A figure is added.

A small section of DENV is missing

  • DENV is added throughout the manuscript

Line 133-140 letters are shown in gray

  • The gray font is converted to black.

Round 2

Reviewer 2 Report

After the revisions, the manuscript underwent considerable improvement.